# Comparative Analysis of Structural Composition and Function of Intestinal Microbiota between Chinese Indigenous Laiwu Pigs and Commercial DLY Pigs

**DOI:** 10.3390/vetsci10080524

**Published:** 2023-08-16

**Authors:** Chao Li, Xueyan Zhao, Guisheng Zhao, Haipeng Xue, Yanping Wang, Yifan Ren, Jingxuan Li, Huaizhong Wang, Jiying Wang, Qinye Song

**Affiliations:** 1Hebei Veterinary Biotechnology Innovation Center, College of Veterinary Medicine, Hebei Agricultural University, Baoding 071000, China; 18253829358@163.com; 2Shandong Key Laboratory of Animal Disease Control and Breeding, Institute of Animal Science and Veterinary Medicine, Shandong Academy of Agricultural Sciences, Jinan 250100, China; 3Key Laboratory of Livestock and Poultry Multi-Omics of MARA, Jinan 250100, China; 4Jinan Animal Husbandry Technology Promotion Station, Jinan 250100, China

**Keywords:** intestinal microbiota, 16S rRNA sequencing, Laiwu pigs, DLY pigs

## Abstract

**Simple Summary:**

The intestinal microbiota produces essential products as well as forms a barrier against pathogens, which has an important impact on pig phenotypes. Recent studies mainly focused on the microbiota of the feces and worldwide farmed commercial pigs, while research on the microbiota of various intestinal sections and indigenous pig breeds is very limited. Laiwu pigs, a precious Chinese indigenous pig breed, are distinguished by their good meat quality, especially their high intramuscular fat (IMF) content, and they also have high resistance to certain infectious diseases. In recent years, people’s demand for meat consumption has transformed from quantity to quality. In this study, intestinal microbiota in the six intestinal segments of Laiwu pigs and the worldwide farmed crossbred, Duroc × Landrace × Yorkshire (DLY) pigs were sequenced by 16S rRNA sequencing and comparatively analyzed to understand the composition and function of microbiota in each intestinal segment, and to explore the influence of intestinal microbiota to phenotypic traits, such as high IMF, high utilization rate of crude fiber, and strong tolerance of Laiwu pigs. This study can help us better understand the microbial characteristics of indigenous pigs and lay a foundation for the potential influence of the host’s genetic background on variation in microbiota composition and diversity.

**Abstract:**

Intestinal microbiota has an important impact on pig phenotypes. Previous studies mainly focused on the microbiota of feces and worldwide farmed commercial pigs, while research on the microbiota of various intestinal sections and indigenous pig breeds is very limited. This study aimed to characterize and compare the biogeography of intestinal microbiota in pigs of one Chinese indigenous breed and one commercial crossbred. In this study, we sequenced the microbiota of six intestinal segments in the grown-up pigs of a Chinese indigenous breed, Laiwu pigs, and the worldwide farmed crossbred Duroc × Landrace × Yorkshire (DLY) pigs by 16S rRNA sequencing, characterized the biogeography of intestinal microbiota, and compared the compositional and functional differences between the two breeds. The results showed that there were obvious differences in microbial structure and abundance between the small and large intestines. Laiwu pigs had higher large intestinal diversity than DLY pigs, while DLY pigs had higher small intestinal diversity than Laiwu pigs. Moreover, some specific bacterial taxa and Kyoto Encyclopedia of Genes and Genomes pathways were found to be related to the high fat deposition and good meat quality of Laiwu pigs and the high growth speed and lean meat rate of DLY pigs. This study provides an insight into the shifts in taxonomic composition, microbial diversity, and functional profile of intestinal microbiota in six intestinal segments of Laiwu and DLY pigs, which would be essential for exploring the potential influence of the host’s genetic background on variation in microbiota composition and diversity.

## 1. Introduction

The intestinal microbiota is a dynamic and enormous ecosystem that produces essential products as well as forms a barrier against pathogens. It plays a pivotal role in morphology, digestion, gene expression, and immunity development [1,2]. It has been estimated that over 170 million unique genes are presented in the human microbiome in the gastrointestinal tract [3]. This number is approximately 8500 times larger than the human gene complement. With the development of high-throughput sequencing technology, metagenome analysis has been widely used to study microbes in the gastrointestinal tract of humans and other mammals [4]. Several studies have demonstrated that a number of human diseases, such as obesity, diabetes, and inflammatory bowel disease [5,6,7], were closely associated with the alterations of gut microbial communities.

Pigs are an important livestock species that not only provide food for human consumption on a large scale but are also closely related to humans, regarding their behavioral patterns, anatomy, physiology, and gut microbiota [8,9]. Thus, they are an excellent model to be used in biomedical studies, including obesity, diabetes, and metabolic disorders [10]. Recent studies have examined the gut microbiome of pigs in relation to their diet [11,12], lipid metabolism [13], developmental stage [14], meat quality [15], antibiotic resistance [16], feed efficiency [17], and growth performance [18].

There are six different segments in the pig’s intestinal tract, including the duodenum, jejunum, ileum, cecum, colon, and rectum. Previous studies have revealed that the microbial structure, composition, and function of different sections of the gastrointestinal tract differed significantly from each other [19,20]. Due to different transit times, pH values, and levels of oxygen and antimicrobials, the small intestine (duodenum, jejunum, and ileum) is dominated by rapidly growing facultative anaerobes such as *Enterobacteriaceae* and *Lactobacteriaceae* [21,22], whereas the large intestine is predominantly occupied by *Prevotellaceae*, *Ruminococcaceae*, *Bacteroidaceae*, *Lachnospiraceae,* and other saccharolytic anaerobes [21,22]. The microbiota in the small intestine primarily regulates the metabolism of simple carbohydrates and amino acids, while those in the large intestine are more conducive to the fermentation of complex polysaccharides [4,21,22]. In most of the previous studies, the microbiota was sampled from feces due to the convenience of sample collection. However, feces only represent the end product of digestion and fermentation processes in the gut and cannot provide a comprehensive view of the colonization of bacteria along the whole gastrointestinal tract. Therefore, studies measuring the gut microbiota and animal traits should use the fecal microbiota with caution [23].

Laiwu pigs are a well-known indigenous breed of Shandong province in eastern China, with a slow growth rate but a high propensity for intramuscular fat deposition [24], while the Duroc × Landrace × Yorkshire (DLY) pigs are a lean, fast-growing crossbred of commercial swine selected for high carcass yield. To the best of our knowledge, we sequenced the microbiota of six intestinal segments in the grown-up pigs of these two populations for the first time by 16S ribosomal RNA gene high-throughput sequencing (16S rRNA sequencing) to characterize and compare the biogeography of intestinal microbiota in these two pig populations with distinct phenotypes. This study provides insight into the shifts in gut microbial diversity, taxonomic composition, and functional profile of Laiwu and DLY pigs, which would be essential for exploring the potential influence of the host’s genetic background on the variation in microbiota composition and diversity.

## 2. Materials and Methods

### 2.1. Animals Information and Sample Collection

A total of 12 castrated about 8-month-old male pigs (Laiwu pigs, *n* = 6; DLY pigs, *n* = 6) were used in this study, and they came from the stock farm of Laiwu pigs and a commercial pig farm adjacent to it, respectively. Laiwu and DLY pigs were fed with fodder formulated according to lard-type pigs and lean-type pigs of the Chinese national standard GB/T 39235-2020 “nutrient requirement of swine” in house feeding, respectively. The diets and their components are detailed in Table 1. All pigs used in this study were healthy and had no diarrhea.

All experimental pigs were electrically stunned and slaughtered after 24 h fasting but with free access to water. Intestinal contents, i.e., digesta, were sampled from different intestinal segments, including the duodenum, distal jejunum, ileum, bottom of the cecum, colon, and rectum, frozen in sterile containers, and stored at a temperature of −80 °C.

### 2.2. DNA Extraction and 16S rRNA Sequencing

Total genomic DNA was extracted from each intestinal digesta mentioned above using Magnetic Soil and Stool DNA Kit (Qiagen, Valencia, CA, USA) according to the manufacturer’s instructions. The concentration and purity of the extracted DNA were measured using NanoDrop ND-2000 spectrophotometer (Thermo Scientific, Waltham, MA, USA), and the integrity was evaluated on 1% (*w*/*v*) agarose gel.

V3-V4 variable region of the 16S rRNA gene was amplified with specific primers of 341F (CCTAYGGGRBGCASCAG) and 806R (GGACTACNNGGGTATCTAAT). The cycling conditions of PCR reactions were as follows: 98 °C for 1 min; 30 cycles of 98 °C for 10 s, 50 °C for 30 s, 72 °C for 30 s, and a final extension at 72 °C for 5 min. After detection on a 2% agarose gel, the PCR products were purified from the gels with Universal DNA Gel Extraction Kit (TianGen, Beijing, China). The purified PCR products were quantified to the same amount to generate the libraries with NEB-Next^®^ Ultra DNA Library Prep Kit (Illumina, San Diego, CA, USA). The libraries were quantified with Fragment Analyzer 5400 (Agilent Technologies, Santa Clara, CA, USA) and sequenced by Novogene Company Limited (Beijing, China) using the PE250 mode of NovaSeq 6000 (Illumina, San Diego, CA, USA).

### 2.3. Data Analysis

#### 2.3.1. Operational Taxonomic Units (OTUs) Annotation

Raw tags for each sample were obtained by cutting barcode and primer sequences and splicing the reads by FLASH (V1.2.7, http://ccb.jhu.edu/software/FLASH/) (accessed on 22 March 2021) [25]. The raw tags were put under strict filtering to obtain clean tags [26], including removing adaptor sequences, empty tags, low-quality tags, and so on. Then, the clean tags were compared with the species annotation database (https://github.com/torognes/vsearch/) (accessed on 22 March 2021) [27] to detect chimera sequences. Finally, the chimera sequences were removed to get effective tags. The effective tags for each sample were clustered into OTUs with 97% identity by UPARSE (V7.0.1001, http://www.drive5.com/uparse/) (accessed on 22 March 2021) [28]. The sequences that were most frequent among the OTUs were selected as representative sequences of OTUs and annotated by species with Mothur (v1.27.0) [29] and the SSUrRNA database of SILVA 138 (http://www.arb-silva.de/) (accessed on 23 March 2021) [30]. We obtained taxonomic information on the community composition and the classification level of the kingdom, phylum, class, order, family, genus, and species with a confidence threshold of 0.8–1.0. The phylogeny of all OTUs representative sequences was obtained by MUSCLE (Version 3.8.31, http://www.drive5.com/muscle/) (accessed on 23 March 2021) [31]. Finally, all data were homogenized using the standard sample that contained the least amount of data. The subsequent alpha and beta diversity analyses were performed based on the homogenized data.

#### 2.3.2. Alpha and Beta Diversity Analyses

To estimate the alpha diversity of each segment of the two populations studied, the Shannon index was computed by QIIME 2 [32] and displayed with the R software. To estimate the dissimilarity in the community structure, Bray–Curtis distances were calculated by QIIME 2 and visualized using principal coordinates analysis (PCoA).

#### 2.3.3. Microbial Community Structure and Differences

To analyze the differences in the composition of the microbiota, the linear discriminant analysis (LDA) effect size (LEfSe) [33], a statistical test that emphasizes both statistical significance and biological consistency was performed with an LDA score threshold > 4.0 as significantly different OTUs between groups.

#### 2.3.4. Function Prediction

PICRUSt2 (https://github.com/picrust/picrust2) (accessed on 28 November 2021) [34] was used to predict intestinal microbiota functions. STAMP [35] was used to compare the differences between groups based on the prediction of genes and their functional characteristics after aligning these with the Kyoto Encyclopedia of Genes and Genomes (KEGG) database (http://kiwi.cs.dal.ca/Software/STAMP) (accessed on 28 November 2021).

#### 2.3.5. Statistics Analysis

The obtained results were represented as the mean ± SD. Statistical analysis was performed using GraphPad Prism version 9.0 (San Diego, CA, USA). A two-tailed Student’s *t*-test or Wilcoxon rank sum test was employed to assess statistical significance. A significance level of *p* < 0.05 was considered statistically significant.

## 3. Results

### 3.1. Summary Statistics for the 16S rRNA Gene Sequencing

In the study, we sampled the digesta of six intestinal segments (duodenum, jejunum, ileum, cecum, colon, and rectum) from Laiwu and DLY pigs. Except for five digesta samples obtained for the jejunum of the DLY pigs, all the other intestinal segments were the six ones. We sequenced the V3–V4 region of the 16S rRNA gene using 16S rRNA sequencing. The biological information of the pigs studied is listed in Appendix A. In total, we obtained 7,005,164 raw paired-end (PE) reads, with an average of 98,664 per sample. After splicing the raw reads to filter raw tags and removing the chimera sequence, a total of 5,836,055 high-quality clean tags were obtained, with an average of 82,198 clean tags per sample. The average length of tags was 410 bp and the average values of Q20 and Q30 were 98.43% and 94.91%, respectively (Appendix A).

To reflect the species richness and evenness of the sequencing data as well as the rationality of the sequencing process, rarefaction curves of OTUs and the Shannon index were calculated with normalized reads to 32,590 for each sample. As shown in Figure 1, the Shannon index and rarefaction curves tend to be flat at the sequencing depth of 32,590, indicating that the amount of sequencing data in the study was sufficient, and more sequences would have resulted in the increase of a limited number of OTUs.

### 3.2. General Comparison of Intestinal Microbiota in Laiwu and DLY Pigs

Based on 97% sequence similarity, all the sequences of the V3–V4 region were clustered into 10,583 bacterial OTUs. To reduce the impact of low-abundant OTUs on subsequent statistical analysis, OTUs with sequence numbers ≥ 2 in at least one sample were retained, and a total of 4217 OTUs were obtained. There were 193 core OTUs detected in bacterial communities in the two populations (Figure 2A). Moreover, 290 and 263 core OTUs were observed in the Laiwu and DLY pigs, respectively (Figure 2B,C).

Shannon’s diversity index measures both the species richness and evenness, and it was used to compare the diversity difference of the microbiota among the six segments and between the Laiwu and DLY pigs. As illustrated in Figure 3, an obvious difference existed between the two populations and among the six segments. Except for the duodenum of the DLY pigs, the large intestinal segments had a more diverse bacterial community than the small intestinal segments, with the duodenum (6.112) and rectum (6.032) showing the highest Shannon index in the small intestinal segments and large intestinal segments, respectively. When we compared the two populations, the Laiwu pigs had higher large intestinal diversity (5.911 on average) than the DLY pigs (5.426 on average), while the DLY pigs had higher small intestinal diversity (4.333 on average) than the Laiwu pigs (4.009 on average). Moreover, the Shannon index difference of duodenum reached a significant level at *p* < 0.01 between the two populations.

Bacterial community structures, analyzed using PCoA based on Bray–Curtis distances, are presented in Figure 4. Samples of the same intestinal segments were basically gathered, demonstrating obvious clustering differences among intestinal segments. In addition, there were obvious differences in species structure and abundance between the small and large intestines. Samples of small intestinal segments were clustered primarily on the left side of the abscissa and those of large intestinal segments were mainly on the right side. Between the two populations, samples of large intestinal segments of the Laiwu pigs overlapped more closely than those of the DLY pigs (Figure 4). This may indicate the functional similarity of the large intestinal segment of the Laiwu pigs.

### 3.3. Analysis of Microbial Composition and Structure

About 98.11% of the OTUs identified (4137/4217) were annotated in the SILVA database. The annotation rates were 94.41%, 89.97%, 83.78%, 70.17%, 49.09%, and 13.54% at phylum, class, order, family, genus, and species level, respectively (Appendix A).

At the phylum level, the bacterial taxa varied greatly among segments of the intestine (Appendix A, Figure 5A,B). The most predominant phylum was *Firmicutes*, which represented 56.13–81.62% and 63.60–89.63% of bacteria of intestinal segments in both Laiwu and DLY pigs, respectively. For Laiwu pigs, the second most abundant phylum was *Proteobacteria* in the duodenum (8.93%), jejunum (15.88%), ileum (35.66%), and cecum (6.78%), and *Bacteroidetes* in the colon (11.16%) and rectum (15.18%). For DLY pigs, the second most abundant phylum was *Actinobacteria* in the duodenum (12.87%), *Proteobacteria* in the jejunum (10.47%) and ileum (7.48%), and *Bacteroidetes* in the cecum (4.79%), colon (9.88%), and rectum (27.44%), respectively. As for the other phyla, they were relatively minor in proportion, representing generally <5% of the bacterial populations throughout the intestinal segment of both Laiwu and DLY pigs. The comparison between the two populations showed that the Laiwu pigs had significantly higher *Proteobacteria* in the ileum and *Euryarchaeota* in the colon and rectum than the DLY pigs (*p* < 0.01), while the DLY pigs had significantly higher *Firmicutes* in the ileum and *Actinobacteria* in the duodenum (*p* < 0.01).

At the genus level, bacterial taxa were quite different throughout the intestinal segments between the Laiwu and DLY pigs (Appendix A, Figure 5C,D). The most predominant genera were *Clostridium_sensu_stricto_1* in the duodenum (25.10%), *Terrisporobacter* in the jejunum (47.60%), cecum (31.14%), colon (16.21%), and rectum (14.72%), and *Escherichia-Shigellain* in the ileum (33.07%) for Laiwu pigs, while they were *Lactobacillus* in the duodenum (20.42%), *Terrisporobacter* in the jejunum (45.37%), ileum (34.91%), cecum (32.50%), and colon (15.64%), and *Streptococcus* in the rectum (12.31%) for DLY pigs. The Laiwu pigs had significantly higher *Terrisporobacter* in the duodenum and rectum than in the corresponding parts of the DLY pigs (*p* < 0.05), while the DLY pigs had significantly higher *Terrisporobacter* in the ileum than that of the Laiwu pigs (*p* < 0.05). Moreover, *Streptococcus* was much more dominant in the colon and rectum of the DLY pigs than in the corresponding segments of the Laiwu pigs (12.31–15.50% vs. <1%), and *Methanobrevibacter* constituted 4.18–4.75% of all bacteria in the colon and rectum of the Laiwu pigs, while it was virtually absent (0.03–0.17%) in the same segments of the DLY pigs.

### 3.4. Bacterial Taxa Differentially Represented in Laiwu and DLY Pigs

To analyze differences in the microbial composition, specific bacterial taxa were identified based on the logarithmic LDA score of 4.0 by LEfSe. As a result, a number of specific bacterial taxa (Figure 6) were found to be differentially represented in all but the jejunum between Laiwu and DLY pigs, demonstrating that many uniquely enriched taxa existed in the specific intestinal segment of the two populations. For instance, 32 differentially represented taxa existed in the duodenum, with 11 and 21 for Laiwu and DLY pigs, respectively (Figure 6A). Moreover, although the most specific taxa were detected in certain specific intestinal segments, some were found in more than one segment. For example, *Lactobacillus* and *Streptococcus* were over-represented in the duodenum, colon, and rectum of DLY pigs (Figure 6A,D,E). A few specific taxa showed a different pattern of dominance between the two populations. For example, *Turicibacter* was over-represented in the duodenum, colon, and rectum of Laiwu pigs (Figure 6A,D,E), while it became more prevalent in the cecum of DLY pigs (Figure 6C). And *Terrisporobacter* was over-represented in the duodenum of Laiwu pigs (Figure 6A) and it became more prevalent in the ileum of DLY pigs (Figure 6B).

### 3.5. Differences in Predicted Function of Ileal and Colonic Microbiota between Laiwu and DLY Pigs

In order to predict the potential influence of intestinal microbiota on the phenotype of the two groups, intestinal microbiota functions were analyzed, and the functional capacity of bacteria in the ileum and colon were further compared between Laiwu and DLY pigs. As illustrated in Figure 7, some significantly different KEGG pathways were identified. The different KEGG pathways mainly belonged to metabolism, as well as genetic information processing, environmental information processing, and human diseases.

In the ileum, biofilm formation, cationic antimicrobial peptide (CAMP) resistance, lipopolysaccharide biosynthesis, flavone and flavonol biosynthesis, biosynthesis of unsaturated fatty acids, pentose and glucuronate interconversions, linoleic acid metabolism, and alpha-linolenic acid metabolism were significantly enriched in the Laiwu pigs, while insect hormone biosynthesis, histidine metabolism, lysine biosynthesis, fatty acid biosynthesis, and biosynthesis of amino acids were significantly enriched in the DLY pigs (Figure 7A). In the colon, pentose and glucuronate interconversions, ferroptosis, protein processing in the endoplasmic reticulum, protein digestion and absorption, riboflavin metabolism, phenylpropanoid biosynthesis, and various types of N-glycan biosynthesis were significantly enriched in the Laiwu pigs, while peptidoglycan biosynthesis, metabolism of xenobiotics by cytochrome P450, drug metabolism-cytochrome P450, retinol metabolism, selenocompound metabolism, D-Alanine metabolism, xylene degradation, and naphthalene degradation were significantly enriched in DLY pigs. Moreover, some immune-related pathways, such as antigen processing and presentation, Th17 cell differentiation, IL17 signaling pathway, and chemokine signaling pathway, were enriched in Laiwu pigs, suggesting that more genes are associated with the immune response existing in the intestinal bacteria of Laiwu pigs.

## 4. Discussion

The swine intestine harbors a vast ensemble of microbes that play a significant role in pig health. With the rapid development of high-throughput sequencing technologies, a number of recent studies have been conducted using 16S rRNA sequencing to characterize the composition and structure of the swine intestinal microbiota [4,16,20]. These studies have greatly expanded our understanding of how intestinal microbiota influence diverse physical characteristics. Nevertheless, the focus of these studies was mainly on the microbiome of feces and worldwide farmed commercial pigs, while they rarely examined the microbiome of various intestinal sections and indigenous pig breeds. Herein, we characterized the composition and function of microbiota in six intestinal segments (duodenum, jejunum, ileum, cecum, colon, and rectum) of a Chinese indigenous breed (Laiwu pigs) and the worldwide farmed crossbred (DLY pigs) and acquired a comprehensive understanding of the biogeography of the swine intestinal microbiome.

### 4.1. Comparison of Microbial Composition and Structure among Intestinal Segments

As a result, different microbial compositions and structures were observed between the small and large intestines in both Laiwu and DLY pigs. On average, the large intestine had more microbial alpha diversity than the small intestine (Figure 3). Moreover, when comparing the data from beta diversity analysis, the small and large intestinal segments were clustered separately on the abscissa (Figure 4). This is not surprising when considering the different physiologies, functions, and ecological environments of the different intestinal sections. In particular, the rectum had the highest microbial diversity among all these segments in both Laiwu and DLY pigs. Previous studies on the association between microbiota and target traits were mainly focused on microbes in feces [15,36]. Our results demonstrated that although feces allow simple and multiple samplings from the same pig, they hardly represent microbial action in the small intestine or the other segments of the large intestine. Consistent with our result, early evidence has also provided a similar conclusion [23]. Therefore, a pretest should be conducted to select appropriate intestinal segments according to experimental requirements.

To date, very limited information is available on duodenal microbiome. In this study, we found that the microbial composition and structure in the duodenum differed greatly from those in other intestinal segments, especially in DLY pigs. The duodenum had the highest microbial diversity among the small intestinal sections in Laiwu pigs and the highest microbial diversity among the whole intestinal sections in DLY pigs. Specifically, the duodenum had the highest abundance of *Lactobacillus* among the six segments in both Laiwu and DLY pigs (Figure 5C,D). A meta-analysis by Devin et al. [19] indicated that the stomach contained the highest concentration of *Lactobacillus*. Thus, the high abundance of *Lactobacillus* in the duodenum likely originated from the stomach. In addition, we also found that *Lactobacillus* has the least abundance in the cecum and increased again in the colon, and its abundance in the colon was still relatively high in the DLY pigs (15.37%). Thus, the concentration of *Lactobacillus* may be affected by pH values and levels of oxygen and antimicrobials of the different intestinal sections.

### 4.2. Comparison of Microbial Composition and Structure between Laiwu and DLY Pigs

Different microbial compositions and structures were also observed between Laiwu and DLY pigs. For example, the DLY pigs had a more diverse microbial community in the three small intestinal segments than the Laiwu pigs; especially, their difference in the duodenum reached a significant level (*p* < 0.05). In contrast, the Laiwu pigs harbored more diverse microbial communities in the three large intestinal segments than the DLY pigs. Laiwu and DLY pigs are two types of populations and are formed under different environments, breeding objectives, and selection intensities. Differences in genetics, environment, and nutrition lead to their distinct and unique appearance and production performance as well as the intestinal microbiota of the two populations. Especially, diets strongly affect the gut microbial composition, which is a particularly influential driver of gut microbiome composition [37]. In history, local people used agricultural by-products with high crude fiber to feed Laiwu pigs, which has formed their high adaptability and utilization rate of crude fiber. In this study, the diet for the Laiwu pigs had a higher proportion of crude fiber than that of the DLY pigs (5.0–7.5% vs. 3.0–3.8%) in order to meet their nutrient requirement. Notably, previous studies have shown that a high-fiber diet can promote the diversity of the larger intestine [38], which is in agreement with the higher large intestinal bacterial diversity of Laiwu pigs. Consistent with Laiwu pigs, a previous study on Tibetan pigs also demonstrated the interaction between environmental conditions and the gut microbiome. Tibetan pigs are an indigenous fatty pig breed in high-altitude and cold areas of China and are raised mainly through stabling and half-stabling feeding with a fiber-rich diet [39]. A study of the gut microbiome in Tibetan pigs indicated that Bifidobacteria, Ruminococcaceae, and Family-XIII-AD3011-Group were conducive to improving disease resistance in Tibetan pigs, and *Lactobacillus* and *Solobacterium* were observed to be the main bacterial communities involved in fat deposition in Tibetan pigs [38].

Specific bacterial taxa (Figure 6) were found to be over-represented in Laiwu pigs, and some of them have been proven to play an essential role in energy metabolism and adipose deposition. These taxa may be related to the high percentage of intramuscular fat of Laiwu pigs. For instance, there was a significant enrichment of *Escherichia coli* in the ileum, *Methanobrevibacter* in the rectum, and Clostridiales in the duodenum of Laiwu pigs. Previous mouse model studies have shown that lipopolysaccharide endotoxin from Escherichia coli could induce obese and insulin-resistant phenotypes [40]. *Methanobrevibacter* can use hydrogen and other products to convert carbon dioxide to methane, which plays an important role in energy metabolism and adipose deposition as shown in a germ-free mouse model [41] and humans [42]. *Clostridiales* include many species of fermentation-associated bacteria, leading to the production of SCFAs and ethanol by fermenting indigestible carbohydrates [43], and they have been revealed to be significantly associated with porcine fatness traits [44,45].

On the other hand, we also identified several over-represented bacterial taxa in DLY pigs, which may be involved in the fast growth rate and high feed efficiency of DLY pigs. For instance, there was a significant enrichment of *Eubacterium* in the ileum and cecum and *Lactobacillus* and *Streptococcus* in the duodenum, colon, and rectum of DLY pigs. *Eubacterium* was revealed to be positively correlated with feed efficiency in previous studies [46,47]. *Lactobacillus* are commonly used as probiotics and are found to be enriched in the cecum [48,49] and feces [36] of more feed-efficient pigs across studies, and they are positively correlated with improved feed efficiency [47]. *Streptococcus* was generally considered to be pathogenic and less abundant in more feed-efficient pigs [50,51]. However, conflicting data have also been obtained for *Streptococcus*, which was found to be enriched in the ileum [51] and feces [36] of more feed-efficient pigs. This may be due to their ability to produce lactic acid and antimicrobials, which would provide energy and reduce their potential as pathogens [52].

### 4.3. Functional Difference of Intestinal Microbiota between Laiwu and DLY Pigs

Finally, we compared the functional difference of intestinal microbiota between Laiwu and DLY pigs and identified significantly different KEGG pathways. Of note, biosynthesis of unsaturated fatty acid, linoleic acid metabolism, and alpha-linolenic acid metabolism were significantly upregulated in the ileum of Laiwu pigs. Unsaturated fatty acids, such as linoleic acid and alpha-linolenic acid, are beneficial to human health [53] and were precursors of meat flavor and have an important impact on the formation of unique flavors of different breeds [54,55]. Previous studies have shown that the contents of total unsaturated fatty acids in Laiwu pork were higher than that in DLY pork [56]. Thus, intestinal microbiota may contribute to the high levels of unsaturated fatty acids in Laiwu pigs and deserves more attention in future studies of fat deposition and meat quality. In addition, some immune-related pathways, such as antigen processing and presentation, Th17 cell differentiation, IL17 signaling pathway, and chemokine signaling pathway, were enriched in the Laiwu pigs, suggesting more bacterial taxa harboring genes related to immune response existing in the Laiwu pigs, which is consistent with the high disease resistance of the Laiwu pigs [57]. On the other hand, pathways related to the metabolism and biosynthesis of amino acids, such as histidine metabolism, D-arginine and D-ornithine metabolism, and lysine biosynthesis, were upregulated in the DLY pigs, which may contribute to the high growth speed and lean meat rate of the DLY pigs.

## 5. Conclusions

In summary, we characterized the microbiota of six intestinal segments (duodenum, jejunum, ileum, cecum, colon, and rectum) in grown-up Laiwu and DLY pigs and compared the compositional and functional differences between the two populations. There were obvious differences in microbial structure and abundance between the small and large intestines. Except for the duodenum of DLY pigs, the large intestinal segments had a more diverse bacterial community than the small intestinal segments. The Laiwu pigs had higher large intestinal diversity than the DLY pigs, while the DLY pigs had higher small intestinal diversity than the Laiwu pigs. Specific bacterial taxa and KEGG pathways were found to be over-represented in Laiwu and DLY pigs, and some of them may be related to distinct phenotypes of populations. This study provides a preliminary exploration of the characteristics of the gut microbiota in Laiwu pigs under natural production conditions. Future research and analysis are needed to verify the results of this study by controlling for feeding environment, diet, and body weight (or age).

## Figures and Tables

**Figure 1 vetsci-10-00524-f001:**
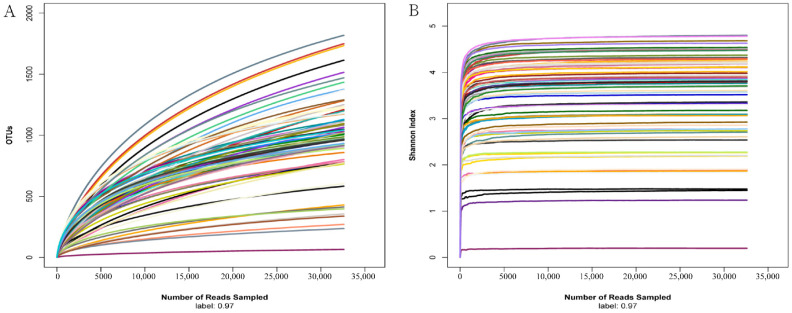
Rarefaction curves of OTUs and the Shannon index for all the samples sequenced in the study. OTUs (**A**) and the Shannon index (**B**) were calculated with reads normalized to 32,590 for six intestinal region samples of all the individuals. Each sample is distinguished by different line colors.

**Figure 2 vetsci-10-00524-f002:**
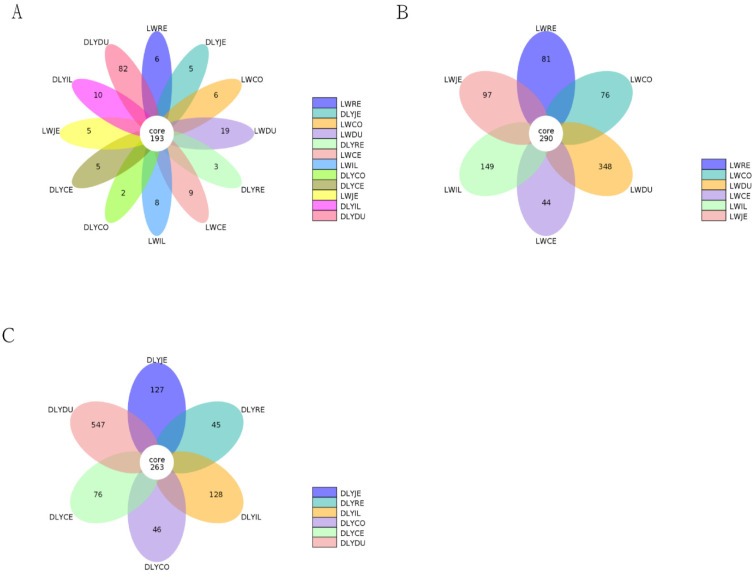
Petal diagram for comparison of the OTU numbers detected in the two breeds. (**A**) Common and unique OTU numbers among different groups in the Laiwu and DLY pigs. (**B**) Common and unique OTU numbers among the 6 groups in the Laiwu pigs. (**C**) The common and unique OTU numbers among the 6 groups in the DLY pigs.

**Figure 3 vetsci-10-00524-f003:**
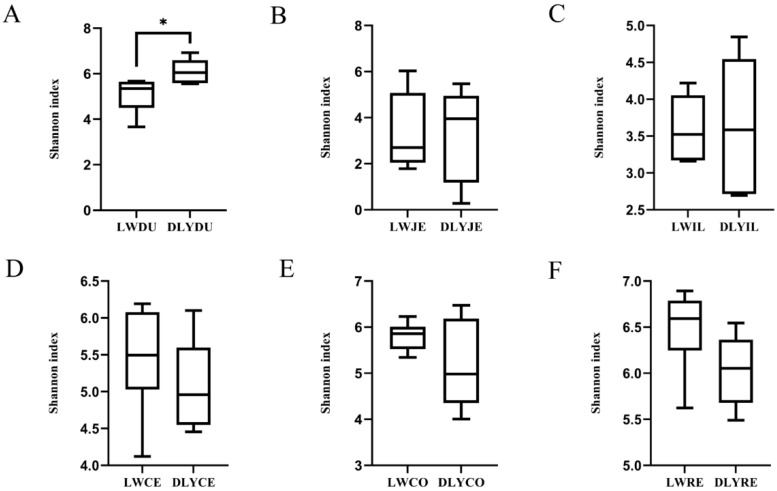
Alpha diversity comparison of the microbiota of Laiwu and DLY pigs based on the Shannon index. (**A**) duodenum, (**B**) jejunum, (**C**) ileum, (**D**) cecaum, (**E**) colon, and (**F**) rectum. Significant mean difference evaluated by Student’s *t*-test are indicated with * for *p* < 0.05.

**Figure 4 vetsci-10-00524-f004:**
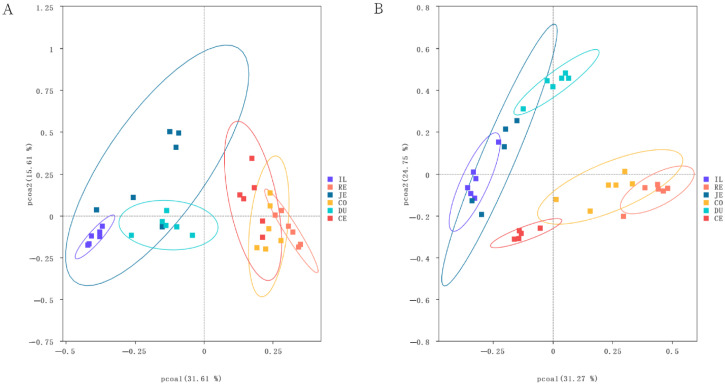
Biogeography of the gut microbiome in Laiwu and DLY pigs. The principal coordinate analysis (PCoA) shows bacterial community structures based on Bray–Curtis distances. On the PCoA plot, each symbol represents one sample. Microbiota in different intestinal segments in Laiwu (**A**) and DLY (**B**) pigs.

**Figure 5 vetsci-10-00524-f005:**
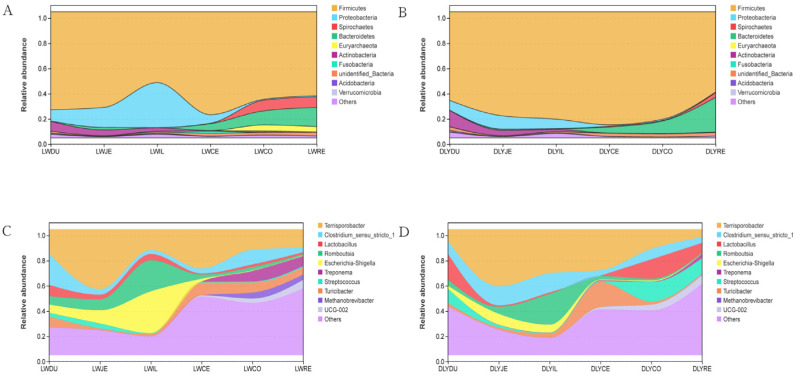
Community composition of the gut microbiota in different intestinal segments of Laiwu and DLY pigs at the phylum and genus levels, respectively. (**A**) Community composition of the gut microbiota of Laiwu pigs at the phylum levels. (**B**) Community composition of the gut microbiota of DLY pigs at the phylum levels. (**C**) Community composition of the gut microbiota of Laiwu pigs at the genus levels. (**D**) Community composition of the gut microbiota of DLY pigs at the genus levels.

**Figure 6 vetsci-10-00524-f006:**
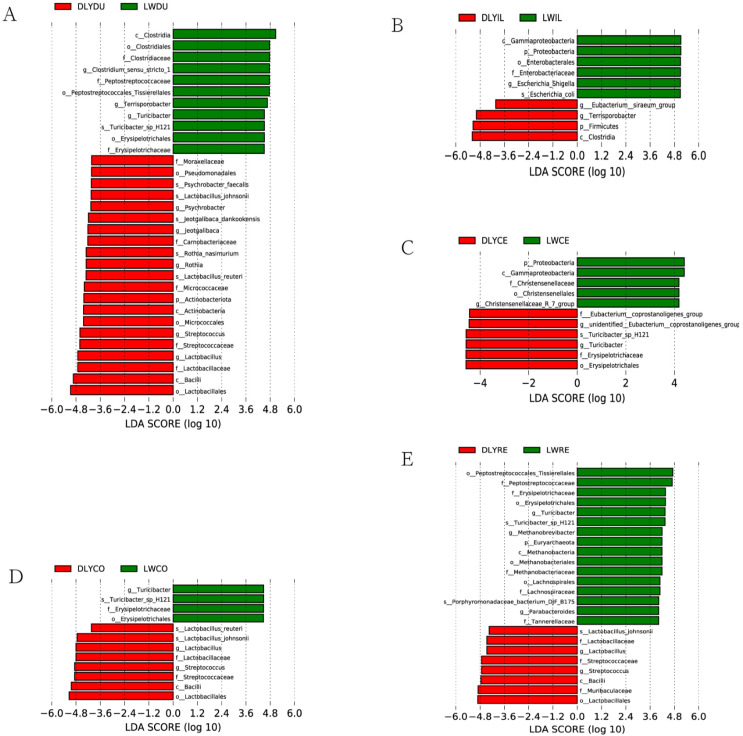
Bacterial taxa differentially represented in the corresponding intestinal segment of Laiwu and DLY pigs identified by LEfSe based on an LDA score threshold of >4.0. (**A**) Bacterial taxa differentially represented in the duodenum. (**B**) Bacterial taxa differentially represented in the ileum. (**C**) Bacterial taxa differentially represented in the cecum. (**D**) Bacterial taxa differentially represented in the colon. (**E**) Bacterial taxa differentially represented in the rectum.

**Figure 7 vetsci-10-00524-f007:**
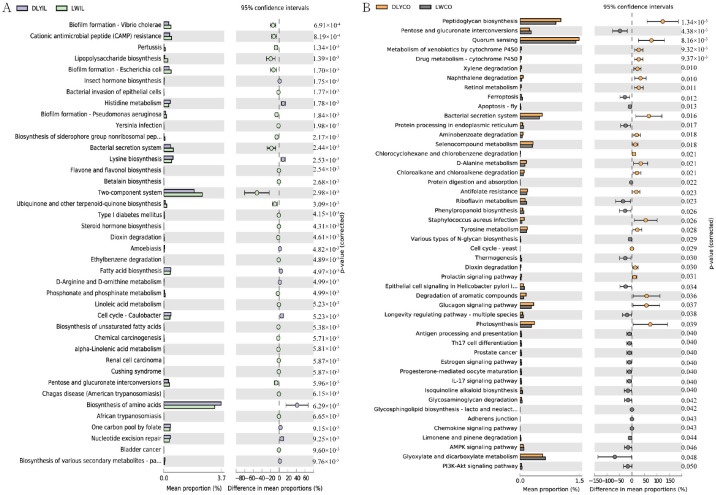
Comparison of predicted function of ileum and colon microbiota between Laiwu and DLY pigs. The significance of the third level of the KEGG pathway between the two groups was performed using the ANOVA test with corrected *p* < 0.05 for the ileum (**A**) and *p* < 0.01 for the colon (**B**).

**Table 1 vetsci-10-00524-t001:** Detailed information on the diets used in the study and their components.

Items	DLY		Laiwu	
Phase/kg	25–60 kg	60–100 kg	25–60 kg	60–100 kg
**Ingredients**				
Corn	64.5	64.0	63.0	56.0
Soybean	21.0	17.0	14.5	9.5
Bran	10.0	14.0	8.0	11.5
Soybean oil	0.5	1.0	2.0	2.0
Peanut Vine	0.0	0.0	8.5	17.0
Concentrate	4.0	4.0	4.0	4.0
Total	100.0	100.0	100.0	100.0
**Nutrient levels**				
Digestible energy (MJ/kg)	14.1	14.0	13.5	12.8
Crude protein (%)	16.1	15.0	13.5	12.0
lysine(%)	0.92	0.79	0.57	0.46
methionine (Met) (%)	0.29	0.23	0.27	0.23
Total calcium (%)	0.63	0.59	0.52	0.48
Available phosphorus (%)	0.18	0.16	0.18	0.16
crude fiber (%)	3.0	3.8	5.0	7.5

## Data Availability

The data presented in this study are available on request from the corresponding author.

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
