# Peer review of "Comparative Analysis of Structural Composition and Function of Intestinal Microbiota between Chinese Indigenous Laiwu Pigs and Commercial DLY Pigs"

_vetsci, 2023, doi:10.3390/vetsci10080524_

Round 1

Reviewer 1 Report

Article vetsci-2482527. Comparative analysis on structural composition and function 2 of intestinal microbiota between Chinese indigenous Laiwu 3 pigs and commercial DLY pigs

Brief abstract: study reported a comparative analysis in terms of regionalization (duodenum, jejunum, ileum, cecum, rectum and colon) about the intestinal microbiota between Laiwu (indigenous) and DLY (cross bred) pigs. Data indicated that microbiome diversity was higher in Laiwu than DLY pigs in large intestine while it was higher in DLY than Laiwu in small intestine; the microbiome functional profile was associated to phenotypic traits including high fat deposition and good meat quality in Laiwu pigs or high growth speed and lean meat in DLY pigs.

Highlights: Study provides foundations about the impact of microbiota in meat quality that may appeal the interest in porcine industry. Congratulations for all authors

Comments

Introduction

Lines 72-76. Microbiota analysis in different segments of the intestine and not in feces should be explicitly justified.

Material and Methods

Line 88 Table 1 Clarify please the term “Phase/kg”

Lines 91 (and 148) clarify please the term “digesta”

Line 94 A section devoted to "Sample processing" is advisable to clarify how each segment was regionalized, the length of each segment collected, furthermore to clarify  whether the feces were removed and intestinal segments were  weighed, the type of sample processed prior tryzol: whole mucosa or whole tissue segment ... and so on

Line 145 Section of “Statistics analysis” was omitted

Results

Remove please the embedded tags in all Figures

Line 165 Figure 1 depicts OTUS and Shannon index of individual samples of each intestinal region from both pig types (72 samples)?

Line 191 Figure 3 Comparisons of Shannon index of two pig types should be depicted in six separated figures i.e. each one for du, je, il, cecum, rectum and colon.

Line 272 clarify please why only in ileum and colon

Discussion

Lines 341-342 according to the Figure 6E, microbial diversity was higher in Laiwu than DLY in rectum; check it up please

Conclusions

Limitations and practical implication of this contributions should be explicitly stated 

Reviewer 2 Report

This study compared the intestinal microbiota of Chinese indigenous breed (Laiwu) pigs and commercial crossbred (DLY) pigs. The researchers analyzed six intestinal segments and found distinct microbial differences between the small and large intestines. Laiwu pigs had higher large intestinal diversity, while DLY pigs had higher small intestinal diversity. Specific bacterial taxa and metabolic pathways were associated with fat deposition and meat quality in Laiwu pigs, and growth speed and lean meat rate in DLY pigs. This study highlights the influence of host genetics on intestinal microbiota variation. Although it was interesting, and it could be improved by addressing the following concerns.

Major concerns.

1.     Animals information and sample collection. the pig’s age? All the pigs used in this study are the same age? Have the authors detected the fecal microbiota compositions before the specific dietary, as it could be the original difference between Laiwu and DLY pigs? What is the feeding environment? How long it was last? More details about this section should be stated.

2.     Rarefaction curve. It would be presented by groups instead of each sample, and as it reflects the quality of sequencing, it would be better to present it in Supplementary Information.

3.     It is very difficult to understand Figure 2. The full name of figure legends would be stated in the manuscript. Please use one-way anova for Figure 3, as it couldn’t show all the comparisons between the two groups.

4.     Please show all the microbiota data in one PCoA pot, including Laiwu and DLY pigs.

Minor concerns.

Line 22. Change ‘16S rDNA-Seq’ to ‘16S rRNA sequencing’ in the whole manuscript.

Line 27. KEGG should list the full name.

Line 132. Change ‘Shannon indices were’ to ‘Shannon index was’.

Line 148-150. Please list the exact numbers of each part from the pigs.

Line 151. PE read?

Reviewer 3 Report

The comparative study of the structure and function of gastrointestinal tract of two pig species is presented in this manuscript. Although it is a simple study, it has used robust methodology with some elements of novelty. Results are well presented and discussed. Manuscript will be of interest and adds to existing literature.

Authors acknowledge Laiwu and DLY pigs are farmed under different environmental conditions, and with breeding objectives. Since diet is one of the major influencers of intestinal microbiota - it is desirable that authors provide more in-depth discussion on this aspect. Usefulness of their findings can also be speculated from that perspective.
